# Hearing Loss in Cancer Patients with Skull Base Tumors Undergoing Pencil Beam Scanning Proton Therapy: A Retrospective Cohort Study

**DOI:** 10.3390/cancers14163853

**Published:** 2022-08-09

**Authors:** Barbara Bachtiary, Dorothe Veraguth, Nicolaas Roos, Flurin Pfiffner, Dominic Leiser, Alessia Pica, Marc Walser, Stefanie von Felten, Damien C. Weber

**Affiliations:** 1Center for Proton Therapy, Paul Scherrer Institute, ETH Domain, 5232 Villigen, Switzerland; 2Department of Otorhinolaryngology, Head and Neck Surgery, University Hospital of Zurich, University of Zurich, 8091 Zurich, Switzerland; 3Faculty of Medicine, University of Zurich, 8006 Zurich, Switzerland; 4Department of Biostatistics, Epidemiology, Biostatistics and Prevention Institute, University of Zurich, 8001 Zurich, Switzerland; 5Department of Radiation Oncology, Inselspital, Bern University Hospital, University of Bern, 3010 Bern, Switzerland; 6Department of Radiation Oncology, University Hospital of Zurich, University of Zurich, 8091 Zurich, Switzerland

**Keywords:** pencil-beam-scanning proton therapy (PBS-PT), skull base tumors, head and neck cancer, meningioma, pure tone average (PTA), hearing loss (HL)

## Abstract

**Simple Summary:**

Most patients with skull base tumors require radiation therapy as part of their overall treatment, preferably with protons. However, vital and healthy organs, such as the cochlea, are often located in the immediate anatomical vicinity of the tumor. Despite the high precision of the proton beam, irradiating the cochlea is often unavoidable, resulting in an increased risk of hearing loss. To assess the frequency and severity of changes in hearing after proton therapy, we performed a retrospective study in a cohort of 51 patients undergoing proton therapy for skull base tumors. We observed that a hearing threshold shift correlates to the applied radiation dose intensity to the cochlea. In addition, advancing age, hearing sensitivity before proton therapy, and the time elapsed after the end of proton therapy are independently associated with the deterioration of the hearing threshold after proton therapy. These results are essential to adequately inform patients about the treatment’s impact and side effects.

**Abstract:**

To assess the incidence and severity of changes in hearing threshold in patients undergoing high-dose pencil-beam-scanning proton therapy (PBS-PT). This retrospective cohort study included fifty-one patients (median 50 years (range, 13–68)) treated with PBS-PT for skull base tumors. No chemotherapy was delivered. Pure tone averages (PTAs)were determined before (baseline) and after PBS-PT as the average hearing thresholds at frequencies of 0.5, 1, 2, and 4 kHz. Hearing changes were calculated as PTA differences between pre-and post-PBS-PT. A linear mixed-effects model was used to assess the relationship between the PTA at the follow-up and the baseline, the cochlea radiation dose intensity, the increased age, and the years after PBS-PT. Included patients were treated for chordoma (n = 24), chondrosarcoma (n = 9), head and neck tumors (n = 9), or meningioma (n = 3), with a mean tumor dose of 71.1 Gy (RBE) (range, 52.0–77.8), and a mean dose of 37 Gy (RBE) (range, 0.0–72.7) was delivered to the cochleas. The median time to the first follow-up was 11 months (IQR, 5.5–33.7). The PTA increased from a median of 15 dB (IQR 10.0–25) at the baseline to 23.8 (IQR 11.3–46.3) at the first follow-up. In the linear mixed-effect model, the baseline PTA (estimate 0.80, 95%CI 0.64 to 0.96, *p* ≤ 0.001), patient’s age (0.30, 0.03 to 0.57, *p* = 0.029), follow-up time (2.07, 0.92 to 3.23, *p* ≤ 0.001), and mean cochlear dose in Gy (RBE) (0.34, 0.21 to 0.46, *p* ≤ 0.001) were all significantly associated with an increase in PTA at follow-up. The applied cochlear dose and baseline PTA, age, and time after treatment were significantly associated with hearing loss after proton therapy.

## 1. Introduction

With radiation therapy, skull base tumors, head and neck cancers, and brain neoplasms are challenging to manage, as many critical organs at risk (OARs) are directly near the target volume [1]. For the former tumors and most of the two other localizations, proton therapy may be the optimal therapeutic strategy if radiation is needed because of the proton doses’ characteristic property (i.e., depth–dose distribution of proton beams, which deposit the maximum dose at the end of their range (in the Bragg peak), followed by a very steep dose fall-off [2,3,4]). As such, protons can deliver a high radiation dose to the tumor while optimally sparing critical OARs to reduce the risk of radiation-induced adverse effects. Consequently, protons have an advantage over photons in that a smaller dose is delivered to the cochlea, although this does not necessarily result in less hearing toxicity [5]. 

Radiation-induced hearing loss (RIHL) is a severe adverse effect that significantly decreases the affected patient’s quality of life [6,7]. The exact pathomechanism of RIHL is not entirely understood. Nevertheless, direct and indirect radiation effects are assumed to cause DNA damage to the hair cells, vascular stria, endothelial cells, and the cochlear spiral ganglia [8]. Overall, cochlear damage related to irradiation is complex, especially in the higher frequencies, and a clear protective cut-off value can hardly be established [9].

Most reports on RIHL came from patients treated with photons and indicated an increase, with the total dose >45 Gy for the fractionated photon radiation therapy [10,11,12]. However, advanced age, impaired baseline hearing level, and ototoxic chemotherapy also raise the risk of RIHL [10,11,12,13]. There is evidence that the characteristics of radiation-induced RIHL differ from chemo-radiation-induced RIHL, mainly in the threshold radiation dose, severity and frequency of RIHL, and the timing of the incident [14].

Although proton therapy is increasingly used worldwide to treat skull base tumors, there is little data on its impact on hearing. This retrospective cohort study aims to assess the frequency and severity of changes in hearing sensitivity after proton therapy of the skull base and to assess potential risk factors for RIHL.

## 2. Materials and Methods

### 2.1. Study Design and Patient Population

This is a retrospective cohort study on patients treated with high-dose pencil-beam-scanning proton therapy (PBS-PT) for skull base tumors at the Paul Scherrer Institute, Switzerland (PSI). The Cantonal Ethics Review board approved the study (EKNZ, 2018-01396). 

From January 2003 to December 2017, 460 patients with a minimum age of 13 years were treated with PBS-PT for skull base tumors. Of this cohort, 51 patients had at least one pre- and one post-treatment audiometry test and were included in the present analysis (Figure 1). 

### 2.2. Treatment 

Technical aspects of PBS-PT planning have been previously described [4,15,16]. According to international recommendations, the treating radiation oncologist contoured the cochlea in the bone window of the planning computer tomography (CT) as a risk organ [17]. If the cochlea was not affected by the tumor, it was excluded from the clinical target volume (CTV) but not from the planning target volume (PTV). For the present study, all cochlear contours were reviewed to ensure they were standardized. The mean dose (Dmean) and the maximum dose (Dmax) applied to the cochlea were used for analysis. 

The dose prescription was performed according to the type of tumor, ranging from 52.2 to 77.8 Gy (RBE) (mean 71.1) in 1.8–2.0 Gy (RBE) daily fractions. A generic RBE factor of 1.1 relative to Co-60 was applied, and the dose was expressed in terms of Gy (RBE). None of the patients in this cohort received concomitant chemotherapy.

PBS-PT was performed using the pencil beam scanning technique at two scanning gantries at PSI. Single-field uniform dose (SFUD) plans and intensity-modulated proton therapy (IMPT) plans were optimized to achieve the prescribed dose in the tumor while covering at least 95% of the PTV and respecting the dose constraints to the organs at risk (OARs). The dose limit at the cochlea was prescribed with Dmean < 45 Gy, and it was spared as much as possible without reducing target coverage.

However, no sparing could be performed in cases where the cochlea was within the CTV or the GTV. In this case, the maximum effort was made to keep the dose to the opposite ear as low as possible. 

### 2.3. Follow-Up Evaluation

Patients were followed with an MRI and a CT at 3- to 6-month intervals in the first 2–3 years after PBS-PT and annually after that. An annual audiometric test was recommended following PBS-PT and was organized at the discretion of the referring centers. 

The audiogram results were reviewed and evaluated with two audiologists for all patients included in this study. The evaluated hearing tests before and after proton therapy are the basis for the present study.

### 2.4. Assessment of Hearing

All patients had a bilateral pure-tone audiometry before starting PBS-PT and at least one audiometry after PBS-PT. A median of 2 (IQR 1–3, range 1–11 tests) follow-up audiometries were performed. One patient was already deaf in one ear during the baseline audiometry; therefore, this ear was excluded from the analysis.

Pure tone average (PTA) was calculated for each ear as the average over frequencies of 0.5, 1, 2, and 4 kHz. Higher values of PTA indicate inferior hearing, and an increase in PTA over time suggests a worsening of the hearing threshold. 

The baseline PTA was classified according to the Global Burden of Disease Expert Group’s (GBDEG) recommendation on hearing loss, in which hearing loss is reported in seven mutually exclusive severity categories [18]. 

The hearing toxicity of proton therapy was classified using the National Cancer Institute Common Terminology Criteria for Adverse Events (CTC-AE) v. 4.03 grading system. In this grading system, hearing loss is reported as any changes at the frequencies of 1, 2, 3, 4, 6, and 8 kHz during follow-up. 

### 2.5. Statistical Analysis

Characteristics of patients and their tumors at the baseline, as well as characteristics of the ears affected by tumor at the baseline, and radiation dose to the cochlea, were summarized as median, interquartile range (IQR), and range for most continuous variables or mean; standard deviation (and range for mean radiation dose to the cochlea); and as frequency and percentage for categorical variables. Characteristics at the first follow-up after treatment, available for all patients, were summarized similarly. The distribution of the frequencies of tumor location (midline, ipsilateral, and contralateral) concerning PTA was analyzed with the Kruskal–Wallis test, the distribution of tumor localization with cochlear dose with the one-way ANOVA, and the relation of tumor localization to hearing impairment and dose group with a chi-square test.

To assess whether the change in PTA between the baseline and the first follow-up (mean 11 months) after treatment was significant, we performed a nonparametric Wilcoxon signed-rank test between the two paired samples.

To assess the association of the radiation dose to the cochlea on the PTA during follow-up, we used the full, longitudinal data of all PTAs on both ears for each patient, including all post-treatment audiometries (range, 1–11) per patient, which were performed at irregular time intervals (overall median follow-up 26 months, IQR 14–69). To account for the hierarchical nature of these data, we used a linear mixed-effects model with a random intercept per ear nested within a patient. PTA at the baseline, patient age in years (at the respective follow-up), years since treatment, and mean dose to the cochlea were used as explanatory variables. To account for the serial autocorrelation between repeated measurements, a continuous, autoregressive process, i.e., an AR (1) process for the continuous-time covariate years since treatment, was used. Age at the follow-up was used instead of age at the baseline to model a separate slope for increasing age and increasing time since treatment, assessing whether the PTA changed more rapidly after treatment than due to increasing age. Coefficient estimates from this model are reported together with 95% confidence intervals (CI) and *p*-values (testing the null hypothesis of each coefficient being zero). Statistical analyses were performed in the R system for statistical computing, version 4.0.4 (R Core Team 2021) [19].

## 3. Results

### 3.1. Patients, Tumors, and Follow-Up

Table 1 summarizes the patient, tumor, and follow-up characteristics. All patients had histologically confirmed chordoma (n = 24, 47.1%), chondrosarcoma (n = 15, 29.4%), head and neck tumors (n = 9, 17.6%), and meningioma (n = 3, 5.9%). None had distant metastases at diagnosis. In 31 patients, the tumor was localized in the midline, and in 20 patients, on one side of the skull base. All patients were treated with curative intent with a mean radiation dose to the tumor of 71.1 Gy (RBE) (range, 52–77.8). The median duration of proton therapy was 51 days (range, 27–60), and the median follow-up time was 26 months (IQR 14–69).

Between the baseline audiometric test and the start of proton therapy, there was a median of 17 days (IQR 8.5–35), and between treatment start to the first follow-up audiometric test, there was a median of 11 months (IQR 5.5–33.7), starting from the beginning of PBS-PT. 

### 3.2. Analysis of Treated Ears

Further analyses were performed on data from all ears of the 51 patients. The one deaf ear at the baseline was excluded, and a total of 101 ears (cochleas) were therefore analyzed.

The median baseline PTA for all ears was 15 dB (IQR 10–25), and grading according to the GBD classification revealed that 17/101 ears (16.9%) already had moderate to profoundly poor hearing (35–95 dB) (Table 2).

The overall mean cochlea dose for all ears was 37.1 Gy (RBE) (SD 22.5). Patients with unilaterally localized tumors had a significantly higher mean dose on the ipsilateral cochlea (59.4 Gy (RBE), SD 16.4) than on the contralateral side (13.4 Gy (RBE); SD 12.29; *p* < 0.001). Additionally, the ipsilateral cochlear dose of lateralized tumors was higher than in both cochleas in midline tumors (59.4 Gy vs. 37.1. Gy (RBE)) (Figure 2).

The median PTA increased significantly by 8.7 dB from 15 dB HL at the baseline to 23.7 dB HL (IQR 11.3–46.3) at the first follow-up, indicating an impairment of hearing sensitivity (*p* < 0.001). This impairment was more pronounced in the ipsilateral ears of patients with lateralized tumors (32.5 dB HL) than in patients with midline tumors (28.9 dB HL) (Figure 3).

In 82/101 (81.2%) ears, the baseline hearing test also included information on bone conduction, which determined sensorineural (24.5%) and mixed (6.9%) hearing disorders as the most common types. At the first follow-up, the percentage of sensorineural disorders was similar (26.7%) to the baseline, but there were significantly more mixed hearing disorders (15.8%) (*p* = 0.047) (Table 3).

### 3.3. CTC Grade Classification of Hearing Loss at First Follow Up 

According to the CTC classification, 16 patients (31%) had an unchanged hearing sensitivity at the first follow-up visit, 11 (22%) patients presented with mild hearing loss of 15–25 dB (CTCAE Grade 1), and 24 patients (47%) presented with moderate to severe hearing loss of >25 dB (CTCAE Grade ≥ 2), respectively (Table 4). Most hearing losses were in the higher frequency range: 13 patients (13.4%) experienced deterioration in the frequency area of 1–4 kHz and 20 patients (20.6%) in 4–8 kHz. Two patients (6%) lost hearing at all frequencies in one ear. 

### 3.4. CTC Grade Classification of Hearing Loss at First Follow Up 

The linear mixed-effects model (Table 5) estimated a significant association between mean cochlear dose (in Gy) and hearing sensitivity measured as PTA at follow-up, with an increase in PTA of 0.34 (95% CI 0.21 to 0.46) per additional Gy when adjusted for baseline PTA, patient age (years), and follow-up time after PBS-PT (year). Further, higher values of baseline PTA (estimate 0.80, 95%CI 0.64 to 0.96), higher patient age (estimate 0.30, 95%CI 0.03 to 0.57 per year), and a longer time after proton therapy (estimate 2.07, 95% CI 0.92 to 3.23 per year) were also independently associated with an increase in PTA at follow-up.

## 4. Discussion

Our data show that patients requiring PBS-PT for skull base tumors are at significant risk for hearing loss. In addition to the applied radiation dose to the cochlea, we identified baseline PTA, age at follow-up, and time after the end of PBS-PT as independent risk factors for hearing loss after proton therapy. 

The clinical effects of ionizing radiation on hearing have been studied mainly with photons. To the best of our knowledge, this is the first investigation to address the impact of PBS-PT on the hearing function in patients with skull base tumors. 

In our study, a gradual relationship was observed between the applied cochlear dose and the deterioration of hearing sensitivity, measured as PTA: each additional Gy to the cochlea resulted in a 0.34 dB increase in hearing loss. 

Interestingly, the changes in hearing were also seen at doses below 32 Gy (RBE), which is significantly lower than the cochlea dose constraints mentioned in the literature [11,20,21,22,23]. 

Previous studies recommended limiting the cochlea’s mean dose to ≤45 Gy, with conventional fractionation [11,20,23]. In a more recent study by De Marzi et al. on combined photon/proton irradiation of 140 patients with skull base tumors, a toxicity rate of 20% was found at a cochlear dose that was <54 Gy and 45% if the dose was >54 Gy [21]. When irradiating childhood brain tumors, Merchant et al. suggested keeping the dose to the cochlea <32 Gy (RBE) [22]. Because the available studies do not provide a precise threshold below which RIHL can be prevented, the QUANTEC report recommends keeping the dose to the cochlea as low as possible [10]. 

Following these common recommendations, the treatment policy at PSI is always to strive for the dose to the cochlea to be below 45 Gy (RBE). However, if the macroscopic tumor is next to or even abutting the cochlea, it is impossible to maintain this dose constraint, even with advanced PBS-PT techniques. In the present study, this was the case in 20 skull base tumors lateralized to one side, where a median dose to the cochlea of 58.8 Gy (RBE) was applied. However, in these situations, special care was taken to keep the dose to the opposite cochlea as low as possible (mean 13.4 Gy) to preserve hearing function in at least one ear (see Table 3). In patients with centrally located skull base tumors, a median dose of 37.5 Gy (RBE) at the cochlea on both sides could be achieved with PBS-PT. 

We also observed that advancing age leads to a hearing loss of 0.30 dB/year, independent of the radiation dose to the cochlea. This result is in line with a recent study on healthy subjects that indicated a progression of hearing loss of 0.29 and 1.35 dB/year (low and high frequencies) in older adults independent of clinical and socioeconomic factors [24]. In contrast to this study, however, we observed a further independent progression of 2.07 dB HL for each elapsed year after PBS-PT. 

Categorizing adverse events after radiotherapy according to the CTC criteria has become widely established as a measurement method for recording. In the present study, 26% of patients had CTC hearing toxicity grades 1 and 2, and 43% had ≥Grade 3 toxicity. However, CTC classification includes hearing changes across all frequencies, even high frequencies essential for daily speech use in noisy surroundings. Thus, CTC classification reflects the true clinical picture from the patients’ perspective for daily life communication. 

On the contrary, the PTA evaluation uses only those frequencies (0.5, 1, 2, and 4 kHz), which are significant for speech understanding when quiet [25]. In the present study, 20 patients had hearing loss at 6–8 kHz, which is relevant for speech understanding when noisy and was considered toxic in the CTC evaluation but not in the PTA evaluation. 

An exciting aspect of our study is that none of the included patients received chemotherapy. This is where our study differs from others, in which primarily concomitant chemotherapy was given, which is a contributing factor for ototoxicity [11,20].

As with all retrospective analyses, this study has potential limitations, including but not limited to uncontrolled patient selection. We performed multivariate analyses to eliminate potential confounders when estimating the association of the mean dose to the cochlea with hearing sensitivity during follow-up, but some residual confounding likely remains. Additionally, the number of cases is small and might lead to non-representative results for a larger patient cohort.

## 5. Conclusions

In summary, we have shown that the baseline PTA, cochlear dose, years after proton therapy, and age at follow-up have independent effects on hearing loss after PBS-PT. Therefore, we believe it is impossible to define a safe dose to the cochlea that will reliably prevent hearing loss after PBS-PT. This fact should be understandably explained to patients so they are sufficiently informed to give informed consent for radiation. 

## Figures and Tables

**Figure 1 cancers-14-03853-f001:**
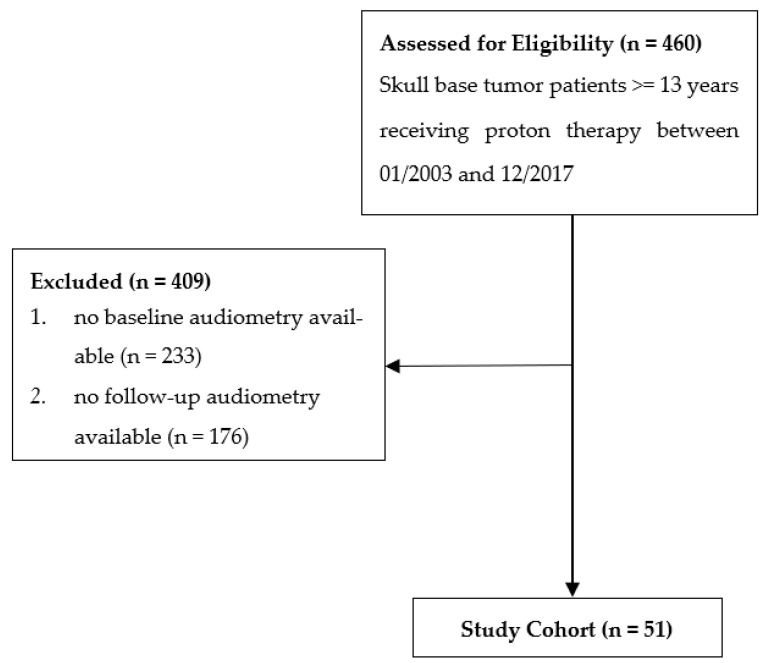
Flow diagram of patient inclusion.

**Figure 2 cancers-14-03853-f002:**
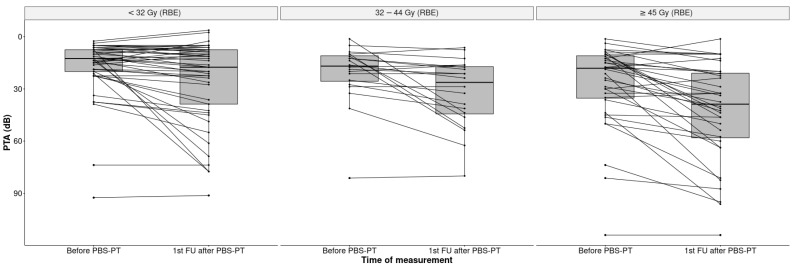
Boxplots of PTA (dB) in all 101 ears before and after PBS-PT according to the cochlear dose in Gy (RBE) (<32, 32–44, ≥45 Gy).

**Figure 3 cancers-14-03853-f003:**
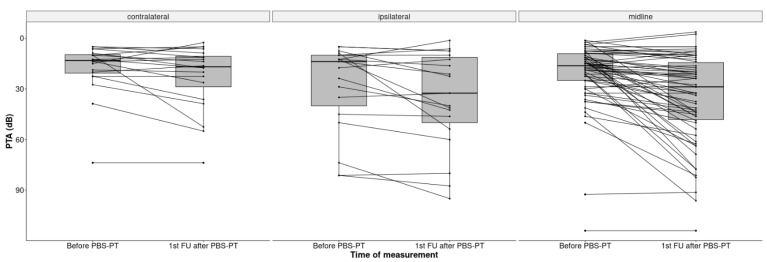
Boxplots of PTA (dB) in all 101 ears before and after PBS-PT according to the tumor position at the skull base (contralateral, ipsilateral, and midline).

**Table 1 cancers-14-03853-t001:** Patients, treatment, and follow-up characteristics (n = 51).

	n = 51
Age at time of proton therapy (median, IQR)	49.7 (39.1–61)
Sex	
- Female, n (%)	30 (58.8)
- Male, n (%)	21 (41.2)
Histology	
- Chordoma, n (%)	24 (47.1)
- Chondrosarcoma, n (%)	15 (29.4)
- Head and Neck Tumor, n (%)	9 (17.6)
- Meningioma, n (%)	3 (5.9)
Tumor position	
- midline, n (%)	31 (60.8)
- lateralized (ipsi and contralateral), n (%)	20 (39.2)
Mean tumor dose, Gy (RBE), mean (range)	71.1 (52–77.8)
Duration or Proton Therapy (days), median (range)	51 (27–60)
Follow-up (months), median (IQR)	26 (14–69)
Number of audiometric tests during follow-up median (IQR)	2 (1–3)
Time interval between audiometric tests	
- Baseline to treatment start (days) median (IQR)	17 (8.5–34)
- Treatment start to first follow-up (months) median (IQR)	11 (5.5–33.7)

**Table 2 cancers-14-03853-t002:** Hearing sensitivity before and after PBS-PT in all 101 ears according to GBD * Expert Group on hearing loss classification.

Hearing Sensitivity (dB)	Before PBS-PT		First Follow-Up	
	n = 101	(%)	n = 101	(%)
Excellent (<5)	5	5.0	4	4.0
Good (5–19.9)	59	58.4	34	33.7
Mild (20–34.9)	20	19.8	23	22.8
Moderate (35–49.9)	9	8.9	17	16.8
Moderately severe (50–64.9)	2	2.0	11	10.9
Severe (65–79.9)	2	2.0	4	3.9
Profound (80–94.9)	3	3.0	5	4.0
Complete (≥95)	1	1.0	3	3.0

* Global Burden of Disease.

**Table 3 cancers-14-03853-t003:** The hearing outcome in the 101 ears of 51 patients.

	Overall	Contralateral	Ipsilateral	Midline	*p*
	n = 101	n = 20	n = 19	n = 62	
Baseline PTA, dB, median (IQR)	15.0 (10.0–25.0)	13.1 (9.7–20.6)	13.8 (10.0–40.0)	16.3 (9.1–25. 0)	0.549
Baseline hearing disorder, n (%)					0.159
conductive	4 (4.0)	0 (0.0)	1 (5.3)	3 (4.8)	
Mixed	7 (6.9)	0 (0.0)	3 (15.8)	4 (6.5)	
normal	46 (45.5)	12 (60.0)	10 (52.6)	24 (38.7)	
sensorineural	25 (24.8)	7 (35.0)	2 (10.5)	16 (25.8)	
unknown	19 (18.8)	1 (5.0)	3 (.15.8)	15 (24.2)	
Follow-up PTA, dB, median (IQR)	23.8 (11.3–46.3)	16.9 (10.6–28.8)	32.5 (11.3–50.0)	28.8 (14.4–48.1)	0.120
Follow-up hearing disorder, n (%)					0.047
conductive	5 (5.0)	0 (0.0)	2 (10.5)	3 (4.8)	
Mixed	16 (15.8)	1 (5.0)	5 (26.3)	10 (16.1)	
normal	27 (26.7)	9 (45.0)	6 (31.6)	12 (19.4)	
sensorineural	27 (26.7)	2 (10.0)	4 (21.1)	21 (33.9)	
unknown	26 (25.7)	8 (40.0)	2 (10.5)	16 (25.8)	
Cochlea Dose Gy (RBE), mean (SD)	36.7 (22.3)	13.4 (12.3)	58.8 (16.7)	37.51 (18.9)	<0.001
Dose Group, n (%)					<0.001
<32 Gy (RBE)	45 (44.6)	17 (85.0)	2 (10.0)	26 (41.9)	
32–44.9 Gy (RBE)	20 (19.8)	3 (15.0)	3 (15.0)	14 (22.6)	
≥45 Gy (RBE)	36 (35.6)	0 (0.0)	14 (73.7)	22 (35.5)	

**Table 4 cancers-14-03853-t004:** CTCAE grade classification at the first follow-up after treatment (n = 51).

CTCAE Grade	Patients, n (%)
0	16 (31)
1	11 (22)
2	2 (4)
3	21 (41)
4	1 (2)

Grade 1: Threshold shift of 15–25 dB averaged at two contiguous test frequencies in at least one ear. Grade 2: Threshold shift of >25 dB averaged at two contiguous test frequencies in at least one ear. Grade 3: Threshold shift of >25 dB averaged at three contiguous test frequencies in at least one ear; therapeutic intervention indicated. Grade 4: Decrease in hearing to profound bilateral loss (absolute threshold >80 dB hearing loss at 3 kHz and above); non-serviceable hearing.

**Table 5 cancers-14-03853-t005:** Effect size estimates and 95% confidence intervals estimated from a linear mixed-effects model on PTA after proton therapy (longitudinal data) with a random intercept per ear nested within patients. The model included 222 audiometric tests on 101 ears of 51 patients. (One patient had a test on only one ear.)

	Estimate	95% CI	t-Value	*p*-Value
PTA before proton therapy (dB)	0.80	0.64–0.96	9.88	<0.0001
Age at follow-up (years)	0.30	0.03–0.57	2.21	0.029
Time since proton therapy (years)	2.07	0.92–3.23	3.57	0.0005
Mean Dose Cochlea (Gy, RBE)	0.34	0.21–0.46	5.43	<0.0001

## Data Availability

The data presented in this study are available upon request from the corresponding author. The data are not publicly available because the PSI repository is currently under construction.

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
