# Peer review of "Hearing Loss in Cancer Patients with Skull Base Tumors Undergoing Pencil Beam Scanning Proton Therapy: A Retrospective Cohort Study"

_cancers, 2022, doi:10.3390/cancers14163853_

Round 1
Reviewer 1 Report
The authors present a large retrospective cohort of skull base tumors treated with pencil beam scanning proton therapy, studying the effect of dose to the cochleae on hearing loss post-treatment. The methods are of high standards and well-described and the study as a whole is of high quality. The conclusions are relevant to the clinical field.
I have a few minor comments:
1. patient inclusion period. The patient inclusion period if from 2003 to 2017. Were there any significant changes on patient selection or patient treatment that could potentially confound with the results? If so, please mention in the methods.
2. in the discussion, the authors rightly indicate that CTCAE grading reflect the clinical impact more than audiograms do. How is the relation between dose to the cochleae and grade =>3, for example? Is there a dose threshold to prevent severe hearing loss?
Author Response
- The Federal Office of Public Health approves the indication for proton therapy of skull base tumors, and the costs for this are therefore covered obligatorily by health insurance. There were no significant changes in patient selection or treatment during the inclusion period from 2003 to 2017.
- We could not find any clear relation between the applied cochlear dose and the CTCAE grades. Also, we could not determine a threshold with which severe hearing loss could be avoided: The mean cochlear dose in the 21 patients with CTCAE grade 3 was 53.7 Gy (RBE); in patients with CTCAE grade 4, the mean dose was 45.4 Gy (Gy), and in patients with CTCAE grade 2, the mean cochlear dose was 50.7 Gy (RBE).
Therefore, we concluded that in addition to the applied cochlear dose, other factors such as age and time after treatment play an important role in hearing loss after proton therapy.
Reviewer 2 Report
An interesting study which adds to the data supporting the use of protons to minimise long-term toxicity.
Well written and clear results and nice that ototoxic chemotherapy was not used. My only comment is that it is a shame that only 11% of the treated patients have adequate data. Are these likely to be the ones most affected? Also no assessment of subjective hearing impairment but I can understand that this would be difficult and objective data is better
Author Response
We agree with the reviewer that it is a pity that audiometry before and after proton therapy was available in only a small proportion of patients.
Unfortunately, it is not possible for us to determine whether the selection included the most affected cases.
In the future, we will consistently require hearing tests from all of our skull base patients before and after treatment.
Reviewer 3 Report
The author did a retrospective study on the connection between the hearing loss and the proton therapy dose to cochlea. According to the authors, their work is the first in doing so, and most of previous studies have focused on the photon therapy. I agree with the authors that the sample size is small (n = 51 patients), and their findings may not be representative of a large cohort. Nevertheless, I believe the authors’ work is still indicative. I also agree with the authors that for the cases that if the tumor is close to the cochlea, it is impossible to reduce the dose to cochlea to the clinical goal. I would recommend the acceptance of this paper with a minor revision.
1. Please spell out CI for Confidence interval at its first appearance.
2. Table 1, yes, I know the value in the () is the percentage, but please clearly label them for each item in the column 2.
3. Please re-organize table 4 to make item and its data match. For example, column 1 has grade and n(%), but actually the percentage is listed in column 2. Move “n=51 patients” to caption.
Author Response
- The confidence interval was spelled out when it first appeared (page 4, highlighted in yellow)
- Table 1 was modified as suggested (highlighted in yellow)
- Table 4 was re-organized as suggested (highlighted in yellow).
Reviewer 4 Report
This article explores the parameters that influence hearing loss in patients treated with proton therapy for skull base tumors in a single-center retrospective study.
The relevant factors were: dose delivered, age, baseline hearing before and follow-up time.
The evaluation of hearing toxicity is complex, combining many possible scores: PTA at several frequencies, WRS (word recognition score), combined score as Gardner-Robertson / AAO-HNSCTCAE or even the GBDEG score classifying the PTA score into 7 categories of severity. The collection of data and its analysis, in particular the statistical work, was satisfactory.
We have thereafter a few basic remarks:
Introduction:
It could be noted that cochlear damage related to irradiation is complex, is notably true in high frequencies and that a clear protective cut-off value is hardly defined (Hua 2008,doi:10.1016/j.ijrobp.2008.01.050).
One the proton therapy side, a sentence explaining the benefit of proton therapy in these skull base tumors and the expected superiority over photon treatments might be of interest. Notably aknowledging, on the hearing toxicity side, that a lesser dose to cochlea with protontherapy doesn't always translate to hearing preservation (Paulino, 2018 DOI: 10.1016/j.radonc.2018.01.002).
Material and method
The use of other hearing assessment scales integrating PTA and WRS as Gardner-Robertson or AAO-HNS and their presentation into scattergram as recommended by the AAO-HNS (Gurgel 2012 doi: 10.1177/0194599812458401) could have been of interest.
Overall, a good and well-conducted work that fits well into a literature that shows the difficulty of predicting the potential preservation of hearing in our patients.
Author Response
Thank you for the two references (Hu et al., Paulino et al.), which we have included in the text (page 2, 55-57 & 62-63)
Thanks also for referencing the hearing assessment scales, which combine PTA and WRS. This is a precious suggestion and would be an excellent addition to our manuscript. Unfortunately, we have not collected the Word Recognition Score and cannot make this evaluation.
Reviewer 5 Report
The main research findings of this paper will be important
Author Response
Thank you for your kind and positive comment.